# Modulated Electro-Hyperthermia-Induced Tumor Damage Mechanisms Revealed in Cancer Models

**DOI:** 10.3390/ijms21176270

**Published:** 2020-08-29

**Authors:** Tibor Krenacs, Nora Meggyeshazi, Gertrud Forika, Eva Kiss, Peter Hamar, Tamas Szekely, Tamas Vancsik

**Affiliations:** 1Department of Pathology and Experimental Cancer Research, Semmelweis University, H-1085 Budapest, Hungary; meggyeshazinora@gmail.com (N.M.); forika.gertrud@med.semmelweis-univ.hu (G.F.); szekely.tamas1@med.semmelweis-univ.hu (T.S.); 2Institute of Oncology at 1st Department of Internal Medicine, Semmelweis University, H-1083 Budapest, Hungary; kiss.eva@med.semmelweis-univ.hu; 3Institute of Translational Medicine, Semmelweis University, H-1094 Budapest, Hungary; hamar.peter@med.semmelweis-univ.hu (P.H.); vancsik.tamas@med.semmelweis-univ.hu (T.V.)

**Keywords:** modulated electro-hyperthermia, irreversible cell stress, apoptosis, damage signaling, immunogenic cell death

## Abstract

The benefits of high-fever range hyperthermia have been utilized in medicine from the Ancient Greek culture to the present day. Amplitude-modulated electro-hyperthermia, induced by a 13.56 MHz radiofrequency current (mEHT, or Oncothermia), has been an emerging means of delivering loco-regional clinical hyperthermia as a complementary of radiation-, chemo-, and molecular targeted oncotherapy. This unique treatment exploits the metabolic shift in cancer, resulting in elevated oxidative glycolysis (Warburg effect), ion concentration, and electric conductivity. These promote the enrichment of electric fields and induce heat (controlled at 42 °C), as well as ion fluxes and disequilibrium through tumor cell membrane channels. By now, accumulating preclinical studies using in vitro and in vivo models of different cancer types have revealed details of the mechanism and molecular background of the oncoreductive effects of mEHT monotherapy. These include the induction of DNA double-strand breaks, irreversible heath and cell stress, and programmed cells death; the upregulation of molecular chaperones and damage (DAMP) signaling, which may contribute to a secondary immunogenic tumor cell death. In combination therapies, mEHT proved to be a good chemosensitizer through increasing drug uptake and tumor reductive effects, as well as a good radiosensitizer by downregulating hypoxia-related target genes. Recently, immune stimulation or intratumoral antigen-presenting dendritic cell injection have been able to extend the impact of local mEHT into a systemic “abscopal” effect. The complex network of pathways emerging from the published mEHT experiments has not been overviewed and arranged yet into a framework to reveal links between the pieces of the “puzzle”. In this paper, we review the mEHT-related damage mechanisms published in tumor models, which may allow some geno-/phenotype treatment efficiency correlations to be exploited both in further research and for more rational clinical treatment planning when mEHT is involved in combination therapies.

## 1. Introduction

High fever-range hyperthermia has been successfully used in clinical oncology for complementing radiation-, chemo-, or even molecular-targeted therapies [1,2,3,4]. Loco-regional hyperthermia is thought to inhibit DNA repair enzymes and increase oxygen supply within cancers to sensitize hypoxic tumor cells for γ-irradiation-induced cell death, e.g., in recurrent breast cancer, locally advanced cervical cancer, or head and neck cancer [5,6,7]. Hyperthermia-related elevated blood perfusion also supports higher intra- and peritumoral drug concentrations to improve the efficacy of chemotherapy, e.g., of localized high-risk soft tissue sarcomas [8]. Further rationale of exploiting hyperthermia in multidisciplinary cancer care comes from the fact that the evolutionary fever response stimulates both the innate and adaptive immune responses [9]. 

The modulated electro-hyperthermia generated by 13.56 MHz amplitude-modulated radio-frequency (mEHT; trade name: Oncothermia) has lately been an emerging way of delivering loco-regional clinical hyperthermia with favorable safety and tolerance profiles [10]. Though mEHT has also been used in combination with chemo-, radiation-, or more recently immunotherapy, a growing number of preclinical studies have demonstrated that it can also induce significant tumor damage in monotherapy, primarily through provoking cell stress and regulated cell death response [11,12]. Tumor selectivity of mEHT treatment is based on the metabolic reprogramming in cancer, resulting in its elevated glycose uptake, oxidative glycolysis (Warburg effect), ion concentration, and electric permittivity (conductivity) compared to adjacent normal tissues [13]. This principle has also been exploited in 2-deoxy-2-[18F] fluoro-D-glucose (FDG) uptake-driven positron emission tomography (FDG-PET/CT) imaging [14].

In mEHT treatment, the tumor is embraced by cooled condenser electrodes (capacitive coupling), which focus radiofrequency current primarily onto ions and charged (dipole) molecules in the extracellular space of cancer tissue and cell membranes [15]. Since cell membrane lipid microdomains enriched in cholesterol and sphingolipids (lipid rafts) [16] carry clusters of regulatory membrane proteins and show stronger dipole potential than the rest of the membrane [17], the selective energy absorption may generate hot spots at lipid rafts [18]. These mEHT–cancer interactions can lead to major heat- and cell stress and are likely to affect essential cell functions through growth regulatory [19] and cell death [20] receptors, ion channels and membrane transport proteins [21], and cell–cell interaction-mediating proteins [22]. The synergy between the thermal (temperature-dependent) and the non-thermal (temperature-independent) effects may explain why mEHT treatment can induce similar or stronger cancer cell death at 38 °C to that gained after traditional radiation heating using water bath (WB) at 42 °C [23], or by mEHT at 42 °C to that by WB at 46 °C [24]. However, the same mEHT protocol may provoke differential death pathways and different extents of tumor damage, depending on the inherent genetic and epigenetic makeup of the treated tumors, as it has been shown by many of these experimental studies.

In this paper, we briefly overview the published preclinical data, the major technical details, including the main treatment specific effects of mEHT and their molecular mechanisms by using a range of tumor models. We essentially considered research papers that were published in peer-reviewed journals accessed through PubMed by searching for the terms “oncothermia”, “electrohyperthermia”, or “electro-hyperthermia”, linked or not with the phrase “modulated”. Besides the early direct tumor damage by mEHT, particular interest has recently been focused on its late secondary impact on promoting damage signaling and immune response, which may also support a systemic (abscopal) effect through immunogenic cell death (ICD) [25]. On the basis of published experiments, we have demonstrated some main events in figures gained from mEHT-treated colorectal cancer models. Our purpose is to highlight major damage pathways activated by mEHT in different tumors to be exploited for more efficient combination treatments of this locally targetable, non-invasive modality with traditional and novel targeted oncotherapy regimens. 

## 2. Standardized mEHT Treatment and Protocols

The mEHT treatment in all reviewed tumor models was delivered using LabEHY-100 or -200 (LE100, LE200) instruments (Oncotherm Kft, Budaors, Hungary) equipped with plan-parallel electrodes (Figure 1a). Intratumoral concentration of electric field is facilitated by the Warburg effect-related increased conductivity (Figure 1b). In vivo, tumors were usually grafted into two distant but standard sites of each mouse; one tumor was treated locally the other served for an untreated control and to test systemic treatment effects. mEHT treatment usually started 7–14 days after inoculation where tumor diameters reached 8–15 mm, which was standardized for limited deviations within each study. A single mEHT shot lasted for 30 min (occasionally for 60 min), and temperature in the tumor or in culture supernatants was kept at 42 (±0.5) °C, monitored using optical sensors, and fine-adjusted by slightly modifying input power. Electric power utilized was 0.4–0.7 W by LE200 and 0.7–2 W by LE100 for in vivo treatments, 1–4.7 W by LE200 and 3–8 W by LH100 for 25 mL cell suspensions, and 6–8 W for LE100 using 65 mL flat vertical cuvettes with coverslip cultures. Treatment dose was measured in real time on the basis of power absorption feedback. Subcutaneous temperature under the cooled electrode surface was kept at about 40 °C, and the rectal temperature was about 37 °C. Tumor genotypic data, which may correlate with therapy response, were obtained from the ExPasy database [26]. All animal experiments were performed in accordance with the Animal Research Act, 1985, on the basis of approvals issued by the local or governmental ethical committees.

The 13.56 MHz radiofrequency has been endorsed and widely used for medical applications with safety, under the conditions described, or when using EHY 2000+ for human treatment between 110–130 W with large electrodes, and no risk of damaging normal tissues or interfering with the action potentials of nerves, cardiac muscle, or with any telecommunication instrument has been found [10,15]. Human mEHT therapy (Oncothermia), which is mirrored in preclinical studies, applies nearly an order of magnitude less of energy for comparable effect when compared with other similar techniques. The carrier radiofrequency current of mEHT is amplitude-modulated using a very low frequency (VLF-AM) within the acoustic range (<20 kHz). The power spectral density is inversely proportional to the frequency (*f*) of the signal (1/*f*), called the “pink noise”. The modulation may increase specific absorption rate (SAR) in the extracellular space and will likely do so also in tumor cell membranes, resulting in enhanced tumor damage compared to radiofrequency alone [27]. In mEHT, SAR is measured and its sum calculated as the absorbed electric power (in *kJ*) by the targeted tumor lump mass embraced by the electrodes. The SAR distribution follows the thermal and electromagnetic heterogeneity of the malignant tissue, resulting in higher absorbance in the cancer than in the surrounding normal issue [27,28]. The general feature of tumor destruction using in vivo mEHT models was the extended morphological tumor damage, starting as early as after 24 h in the middle of the treated tumor lamps, which then almost evenly spread from inside to the periphery (Figure 1c) [11,12]. This suggested the instant penetration, concentration, and enhanced energy absorption in the middle of tumors.

### Methods Used for Demonstration Images

Details of the experimental models, including ethical approvals by the National Scientific Ethical Committee on Animal Experimentation (no. SzIE ÁOTK MÁB 233/2012 and no. SzIE, ÁOTK MÁB 26/2013 and the most recent one, no. PE/EA/633-5/2018), using mouse C26 colorectal cancer cell line allografted into BALB/c mice, are found in Vancsik et al., 2018 and 2019 [12,29]. In the studies, 10% neutral-buffered, formalin-fixed tumor tissues embedded in paraffin were cut for 2–3 µm thick serial sections and mounted on silanized adhesion glass slides. Dewaxed sections were rehydrated for hematoxylin–eosin staining and immunohistochemistry. For antigen retrieval, heating at 100 °C for 40 min using a microwave oven (Whirlpool, Benton Harbor, MI, USA) in Tris-EDTA (TE) buffer (pH 9.0; 0.1 M Trisbase and 0.01 M EDTA) was applied, followed by a 20 min cooling. Endogenous peroxidases were blocked using 3% H_2_O_2_ in methanol while the non-specific protein binding sites were blocked in 3% bovine serum albumin (BSA) diluted in 0.1 M Tris-buffered saline (TBS; pH 7.4) containing 0.01% sodium-azide, both for 15 min. Then, sections were incubated with Western blot-validated primary antibodies (Table 1) diluted in 1% BSA/TBS+Tween20 (TBST; pH 7.4) in a humidity chamber overnight. Peroxidase-conjugated anti-rabbit IgG (Histols micropolymer, Histopathology Ltd., Pecs, Hungary) was used for 40 min and the enzyme activity was finally revealed in a 3,3’-diaminobenzidine (DAB) chromogen/hydrogen peroxide kit (Thermo Fischer Scientific, Waltham, MA, USA) under microscopic control. All incubations were at room temperature, with the samples washed between incubations in TBST buffer for 2 × 5 min. 

For immunofluorescence detection, Alexa Fluor 488 (green)- or Alexa Fluor 546 (orange-red)-coupled anti-rabbit Ig (1:200) was used for 90 min. Cell membranes were highlighted with a wheat germ agglutinin (WGA) Alexa Fluor 488 conjugate (1:200), and cell nuclei were stained blue with 4’,6-diamidino-2-phenylindole (DAPI). All fluorescence probes were from Thermo-Invitrogen (Carlsbad, CA, USA).

All antibodies were from rabbit except *F4/80, which was from rat to be converted with and intermediate incubation step using rabbit anti-rat IgG (1:200, **#**31218, Thermo).

Vendor specification: Cell Signaling (Danvers, MA, USA); Thermo (Waltham, MA, USA); Dako (Glostrup, Denmark); Cell Marque, Sigma-Aldrich (Rocklin, CA, USA); T-E: Tris-EDTA, pH 9.0; citrate: 0.1 M citrate-citric acid buffer, pH 6.0.

## 3. mEHT-Induced Apoptosis

Cells are more responsive and sensitive to heat under oxygen-deprived and highly acidic conditions, which are common in cancer tissue due to aggressive growth and enhanced glycolysis [13,30,31]. Despite the diverse thermo-tolerance of different cancers, the common feature of all reviewed mEHT treatments is that they caused significantly higher tumor damage compared either to untreated in vivo (sham) controls (or cell cultures kept at 37 °C) or to traditional heat radiation (WB, or infrared lamp) treatment also using 42 °C for the same duration (Table 2).

Either using human or animal malignant tumor cell lines under in vitro or in vivo settings, the dominant pathway of tumor destruction after mEHT was apoptosis mediated programmed cell death as opposed to necrosis, as demonstrated mostly using annexin V–propidium iodide flow cytometry [29,42]. Apoptosis was also confirmed by the widespread detection of its characteristic morphological signs in hematoxylin–eosin-stained sections or cell cultures, including nuclear shrinkage and chromatin condensation (picnosis), cell membrane blebbing, and formation of apoptotic bodies (Figure 1d) [11,12,18,23]. These features resulted in significantly paler areas of damaged vs. intact-looking tumor parts, as tested at low power view in whole digital slides, which allowed the accurate demarcation, measurement, and comparison of the damaged to the whole tumor area, defined as tumor destruction ratio (TDR). Dividing TDR of the treated by TDR of the distant (untreated) tumor nodules in the same animals defined direct tumor destruction efficiency (TDE) of mEHT. Regulated cell death response was also supported by detecting elevated DNA fragmentation using TUNEL (terminal deoxynucleotidyl transferase dUTP nick end labeling) assay; subG1-phase cell fraction with flow cytometry; and the upregulation and activation of enzymes mediating different pathways of apoptosis using quantitative RT-PCR, RNA sequencing, apoptosis protein arrays, Western blots, and immunohisto-/cytochemistry [11,29,37,43].

### 3.1. Caspase-Dependent Apoptosis

The tumor suppressor TP53 gene is considered to be a major guardian of the genome, which is activated in response to cells stress, including heat and DNA damage [49]. Besides regulating protective responses such as cell cycle arrest, cell senescence, DNA repair, and metabolic adaptation, activated p53 can induce the transcription of genes mediating programmed cell death, e.g., in response to anticancer therapy [50].

C26 (known also as CT26), an aggressive colorectal cancer (CRC) cell line induced into a BALB/c mouse using N-nitroso-N-methylurethane (NMU), offered an ideal treatment model. C26 is characterized by microsatellite stability (MSS) and wild type Tp53 and Apc genes, besides mutant Kras, Cdkn2a-del, hypermethylation, and a 21-nucleotid deletion in the Krt8 gene [51,52]. Krt8-del/met shifts the C26 phenotype towards vimentin expression and supports epithelial–mesenchymal transition (EMT). In C26 culture, a single 30 min shot of mEHT induced the early (3–9 h post-treatment) upregulation of the pro-apoptotic Bax and Puma, and reduced anti-apoptotic Xiap, Bcl-2, and Bcl-xl transcripts [29,51,52]. After 24 h, elevation and nuclear translocation of phospho-p53 (Ser15) and reduced phospho-Akt(Ser473) protein levels were detected, along with a significant caspase-3-mediated apoptosis. In C26 cells grafted into BALB/c mice, pathomorphology and DNA fragmentation also confirmed mEHT-induced massive apoptosis [12]. The significant increase of cleaved/activated caspase-8- and caspase-3-positive tumor cell numbers, the mitochondrial translocation of bax, and the cytoplasmic release of cytochrome-c proteins clearly indicated the activation of both the extrinsic and the mitochondrial caspase-dependent regulated cell death pathways (Figure 1e).

Cleaved caspase-3 is known to activate endonucleases, primarily the caspase-3-activated DNase (CAD), which can cleave nuclear DNA into small internucleosomal fragments of ≈180 base pairs (bp) or its multiplied values [53]. Apoptosis in C26 models was likely to be driven by p53 protein, which was detected in an increased number of tumor cell nuclei, and its phosphorylation at Ser15 was compatible with its stabilization against Mdm2-mediated ubiquitination [54]. Reduced Akt levels after treatment were in line with the inhibition of both mdm2 and the anti-apoptotic Xiap, which could have interfered with caspase-3 activation [55,56]. Therefore, in Tp53 wild type colorectal cancer, the major damage pathway activated by mEHT was caspase-dependent apoptosis involving both the extrinsic and intrinsic subroutines.

In HepG2 hepatocellular carcinoma (hepatoblastoma) carrying wild type TP53 gene (but NRAS p.Gln61Leu c.182A > T mutation), a single shot of mEHT also prompted elevated expression of caspase-3, -8, and -9, leading to a massive apoptosis [24]. Another group also found strong induction of caspase-mediated apoptosis in HepG2 cells after three repetitions of mEHT treatment of 60 min each, where RNA sequencing revealed the upregulation of the proapoptotic and growth suppressor septin 4 (SEP4) as a potential driver, which was confirmed at the protein level [38].

The U87-MG (homozygous PTEN c.209+1G > T, and TERT c.228C > T, -124C > T mutations) and A172 (ABL1–CBFB gene fusion) human glioma cell lines carrying wild type TP53 also reacted with massive apoptosis to mEHT treatment through DNA damage and poly-adenyl ribose polymerase (PARP) downregulation-induced E2F1 stabilization and upregulation [37]. Elevated p53 protein levels suggested the involvement of caspase cascade, although E2F1 can also initiate apoptosis in TP53 mutant tumors [57]. However, activation of E2F1 is a double-edged sword as it can also promote tumorigenesis and progression, depending on the anti- vs. proapoptotic signals received [58].

Interestingly, despite the homozygous TP53 c.559+1G > A (and hemizygous PTEN p.Gly129fs*51, c.387_388insCGCC) mutations, a single shot of mEHT induced elevated cleaved caspase-8 and –caspase-3 and truncated Bid levels 3 h post-treatment and a massive apoptosis later (tested for DNA fragmentation and annexin V labeling) in U937 myelomonocytic lymphoma [18]. This was accompanied by treatment-related significantly reduced mitochondrial membrane potential (damage) and elevated intracellular Ca^2+^ levels, which both were likely to contribute to augmented regulated cell death [59]. Gene expression array revealed the upregulation of the pro-apoptotic tumor necrosis factor receptor superfamily (FAS)- and c-JUN N-terminal kinase (JNK)-mediated pathways with a potential to drive this process [18].

### 3.2. Caspase-Independent Apoptosis

Although the major pathways of apoptosis are regulated by activated TP53, this gene is mutated in about 50% of all solid tumors [49]. Therefore, when this happens, alternative mediators of programmed cell death must be recruited.

The human HT29 CRC cell line was isolated from a Dukes C stage tumor (with fat and lymph node invasion) and proved to be ideal to study mEHT effects on TP53 mutant (homozygous p.Arg273His (c.818G > A) cancer with deregulated canonical apoptosis induction [60]. Additional features of HT29 cells involved MSS, BRAF mutation linked with MAPK pathway activation and therapy resistance [61], and PI3KCA mutation known to activate AKT/mTOR signaling and promote metabolic shift towards glycolysis [62]. In HT29 CRC xenografted into BALB/c nu/nu immunocompromised mice, a single 30 min mEHT shot also provoked significant apoptosis, DNA fragmentation, and formation of apoptotic bodies [9]. Treatment-specific tumor destruction (TDE) increased up to sevenfold by 72 h post-treatment. However, despite the increased cell membrane TRAIL-R2 death receptor [32], a potential mediator of extrinsic apoptosis, as well as of the mitochondrial BAX and cytoplasmic cytochrome c, the promoters of intrinsic programmed cell death, researchers only detected cleaved/activated caspase-3 positivity in some peri- and intratumoral inflammatory cells, but not in tumor cells.

Searching for alternative, caspase-independent pathways revealed the significant mitochondrial release and nuclear translocation of apoptosis inducing factor-1 (AIF1) in tumor cells from the early apoptotic stage [11]. Increased expression of activated/cleaved AIF1 protein of 57 kDa (full size: 62 KDa) was confirmed in mEHT-treated tumors, which could also mediate nuclear chromatin condensation and cleavage of DNA into larger ≈50 kb fragments through the recruitment of nucleases, such as cyclophilin A (CYPA) or endonuclease G [63]. Therefore, in TP53 mutant CRC, the missing functional p53 protein could not trigger caspases upon mEHT-induced irreversible cell stress. Instead, AIF1, released through bax-produced mitochondrial outer membrane pores, became activated and translocated to cell nuclei, promoting DNA fragmentation. Furthermore, mutant p53 is known to enhance metabolic shift (Warburg effect) [64], and thus the electric conductivity and cell stress, which may explain the highly efficient mEHT-induced tumor destruction, despite the impaired canonical apoptosis.

## 4. mEHT-Induced DNA Double-Strand Breaks and p21^waf1^-Mediated Senescence

The transcription factor p53 plays a central role in regulating major cell adaptation responses including the cell cycle control, DNA repair, or apoptosis when faulty DNA cannot be rescued, e.g., in cancer cells [50]. Thus, it is not surprising that this important tumor suppressor shows high mutational prevalence in cancer. Besides recruiting the caspase cascade upon heat and cell stress, p53 activation can also upregulate the cyclin-dependent kinase inhibitor p21^waf1^ (CDKN1A) as it was shown both at mRNA and protein levels between 3–9 h and 24 h post-mEHT treatment, respectively, in C26 cell cultures and also in tumor allograft sections [29]. Significant elevation of p21^waf1^ levels was consistent with the complete cell cycle arrest indicated by the loss of Ki67 positivity in the damaged central region of the highly proliferating C26 CRC [12]. Other tumor models where mEHT treatment induced elevated p53 and p21^waf1^ levels included HepG2 hepatocellular carcinoma cell cultures [38] and B16F10 melanoma cells (with Cdkn2a-del, which downregulates Tp53, and Braf mutations), both in culture and mouse xenografts [42].

Enhanced p21^waf1^ expression can arrest the cell cycle both at G1 and G2 phases to prevent damaged DNA to be duplicated or cells with DNA damages to enter mitosis and let DNA repair mechanisms be recruited [65]. Permanent p21^waf1^ upregulation and cell cycle arrest is called senescence, which may show transitional therapeutic benefits, but hinders radiation-induced tumor destruction [66]. Moreover, incomplete senescence is thought to be used by tumors as an adaptive pathway to delay but restart proliferation and become more aggressive [67]. mEHT treatment significantly reduced tumor growth both in HepG2 xeno- and B16F10 allograft models as tested 1 week after the first (or only a single) treatment [38,42]. Since elimination of tumor debris by phagocytosis takes weeks, senescence also likely to play a role in reduced tumor sizes after mEHT when tested in this time frame. This idea was found to be even more feasible for B16F10 melanoma, where histological signs of apoptosis and the elevation of Puma, Aif1, and activated-caspase-3 levels were only modest [38,42].

Hyperthermic temperatures (41.8 °C or 43 °C) can also induce DNA damage, including double-strand breaks (DSB), as shown in human sarcoma (U2OS–osteosarcoma, SW872–liposarcoma, SW982-synovial sarcoma, RD-ES–Ewing sarcoma, and SKUT-1–leiomyosarcoma) and human colorectal carcinoma (DLD1) cell lines [68]. DSBs are indicated by the phosphorylation of histone 2AX (H2AX) at serine 139 observed, first after *γ*-irradiation, which is why it is marked as *γ*H2AX [47]. In line with conventional hyperthermia, mEHT also provoked the granular upregulation of nuclear *γ*H2ax in significantly elevated numbers of C26 CRC and B16F10 melanoma cells [29,42]. Since DNA fragmentation after mEHT monotherapy had been detected in several tumor models, we had a good reason to suggest that upregulated *γ*H2ax protein indicated DSBs. DNA damage-induced *γ*H2ax is thought to be essential for p21^waf1^ upregulation (through p53 activation), which was detected in mEHT-treated B16F10 and C26 tumor models (Figure 2) [65]. High levels of *γ*H2ax in damaged, non-proliferating tumor cells are likely to be mEHT treatment-related. Interestingly, DSB-induced *γ*H2AX also initiates specific association with activated AIF after stress-related mitochondrial release and nuclear translocation of AIF [63]. *γ*H2AX/AIF then forms a complex with and activates CypA endonuclease for apoptogenic activity.

Besides inducing *γ*H2AX loci, hyperthermia can reduce BRCA2 expression and thus BRCA2-induced recruitment of RAD51 recombinase for repairing DNA [68], and may also interfere with several other DNA repair pathways [69]. This can further explain why hyperthermia in general and mEHT can be efficiently combined with DNA-damaging treatments, including radiotherapy and platinum-based or topoisomerase inhibitor chemotherapeutics, or PARP1 inhibitors, despite also supporting the DSB-induced senescence pathway. Of note, mEHT can directly reduce PARP expression both in OVCAR-3, a high grade serous ovarian adenocarcinoma cell line carrying homozygous TP53 p.Arg248Gln (c.743G > A) mutation [43], and in U87-MG and A172 human glioma cells of wild type TP53 [29,37]. Low PARP levels allow easier single-strand DNA breaks to be accumulated and sensitize DNA for more easily suffering double-strand breaks, particularly when repair enzyme levels such as those of BRCA2 are also reduced by hyperthermia [70].

## 5. mEHT-Induced Cell Stress, Chaperones, and Damage Signaling

Chaperone proteins are upregulated by heat and cell stress, as well as in cancer, where abnormally folded neoantigens are constantly produced [25]. The major chaperones are heat shock proteins, which support the conformational folding and assembly of macromolecules, particularly proteins, to prevent their aggregation and denaturation [71]. Elevated intracellular HSP70 levels protect cancer cells from regulated cell death and support treatment resistance [72]. However, cell membrane bond and extracellular heat shock proteins carry neoantigen fingerprints of tumor cells and promote their uptake by phagocytes and antigen-presenting dendritic cells (DCs) through pattern recognition receptors such as TLR2/4 [25].

Modulated EHT treatment caused significant destruction of HT29 CRC xenografts grown in immunocompromised (BALB/c nu/nu) mice, besides upregulating heath-shock proteins (HSPs) and calreticulin. HT29 CRC showed early and elevated HSP response involving HSP40, HSP60, HSP70, and HSP90 mRNAs 4 h after mEHT treatment [34]. At 14 h, Hsp70 protein was mainly localized to the cytoplasmic membranes in damaged tumor areas, and its expression doubled by 24h post-treatment. It is important to note that in HepG2 cells, mEHT-induced significantly higher HSP70 and calreticulin upregulation and HSP70 release than WB heating or conventional capacitive coupling hyperthermia (cCHT) [24]. The earliest chaperone responding to mEHT stress was calreticulin, which showed increased expression and was cytoplasmic to cell membrane translocation in HT29 tumors as early as 4 h post-treatment [34]. Calreticulin is an endoplasmic reticulum (ER) Ca^2+^-binding chaperone that prevents misfolded proteins to leave the ER [73]. Cytoplasmic release of calreticulin in bulk from the ER is prompted by massive cell stress or damage induced, for instance, by chemotherapy or UVC irradiation and exposed to cancer cell membranes (ecto-calreticulin). This serves as an “eat me” signal, also carrying tumor antigens to phagocytic receptors such as CD91 of macrophages and DCs [74]. The intracellular Ca^2+^ in cancer cells accompanying calreticulin release in HT29 CRC was also likely to promote apoptotic tumor damage [59]. Therefore, mEHT could induce the cell membrane translocation and release of both chaperone proteins in HT29 xenografts to become parts of damage signaling, which can promote the uptake of tumor antigens by innate immune cells, i.e., macrophages and DCs for cross-presentation and activation of anti-tumor T cell response [12,75].

The release of HMGB1 (high mobility group box 1) protein from cell nuclei was also triggered by mEHT 24 h post-treatment and then released and cleared from tumor cells after 48 h [12,34]. HMGB1 is a non-histone chromatin protein that interacts with histones, DNA, and transcription factors and thus regulates transcription, e.g., of TP53 and NF-κB [76]. Massive cell stress can induce its nuclear-to-cytoplasmic release after hyperacetylation on lysine residues, then its bulk extracellular release is dominantly post-apoptotic. Extracellular HMGB1, known as a “danger signal”, also carries neoantigens from tumor cells, including fragments of DNA, to find specific pattern recognition receptors on DCs such as RAGE (receptor for advanced glycation end products) and TLR4 (Toll-like receptor-4, CD284). These results in HT29 CRC suggested that damage signaling relevant for immunogenic cell death (ICD) can be induced by mEHT, even in immunocompromised animals.

## 6. Contribution to Systemic Effect and Secondary Immunogenic Cell Death

The spatiotemporal release of HSP70, calreticulin, HMGB1, and some other molecules, e.g., ATP or S100 proteins from tumor cells, are known as damage-associated molecular pattern (DAMP) signals that can promote the uptake, processing, and presentation of tumor antigens by DCs [74]. This adjuvant mediated antigen uptake can support the maturation of DCs through upregulating major histocompatibility complex (MHC) class I and class II molecules; CD80 and CD86 costimulatory molecules; Th1 cytokines IL-2, IL-12, and IFNα, which can also activate NK-cells; and chemokines for efficient antigen cross-presentation and activation of antitumor T cell response [25,75]. However, it is important to consider that inflammation induced by DAMPs may also contribute to carcinogenesis and progression in a context-, tumor stage-, and tumor-specific manner [74].

Since HT29 CRC was grown in immunocompromised mice, the potential of DAMP damage signaling to induce secondary immune-mediated tumor cell death was tested and confirmed in mouse C26 CRC grafted into immunocompetent BALB/c mice [12]. Of the symmetrical tumors grown in both femoral regions of mice, the mEHT-treated one in the right leg always showed significant tumor damage by apoptosis compared to the untreated left leg tumors and sham controls. The prominent mEHT-related upregulation and cell membrane accumulation of calreticulin after 12 h and Hsp70 after 48 h, accompanied by the nuclear-to-cytoplasmic Hmgb1 translocation and release from tumor cells from 48 h post-treatment, confirmed significant DAMP signaling (Figure 3). Minor treatment-related effects were regularly seen in the untreated left tumors of the same animals compared to the sham controls, calling attention also to a potential systemic effect.

Indeed, mEHT treatment combined with the intraperitoneal injection of a chlorogenic acid-rich T cell-promoting agent (MTE) amplified the tumor damaging effect to significance also in the untreated left tumors distant from the treated sites (abscopal effect) [12,77]. This was accompanied at both sites (compared to sham control) with the tumor infiltration by S100-positive antigen-presenting DCs, CD3, and granzyme-positive T cells; granzyme-positive NK cells; and F4/80-positive macrophages (Figure 4), with only scant FoxP3-positive regulatory T cells [12]. Progressive accumulation of tumor damage and immune response after a single mEHT shot suggests that after a direct, primary apoptosis induction, the combined treatment led to a secondary ICD-mediated tumor damage. The mEHT treatment also induced significantly elevated ROS (reactive oxygen species) and cell membrane bond Hsp70 levels, as well as extracellularly released ATP, calreticulin, and Hmgb1 levels in B16F10 melanoma grafts in C57B1/6 mice [42]. However, these were unable to recruit measurable anti-tumor immune response due to almost missing Mhc-I and reduced Melan A expression.

The ICD is known to be supported by tumor destruction, either by necrosis, apoptosis, or mixed death pathways in response to chemotherapy (e.g., doxorubicin, oxiplatin) [78,79]; radiotherapy [80]; epidermal growth factor receptor (EGFR) targeting cetuximab immunotherapy [81]; or immune checkpoint inhibition therapy [82]. The combination of mEHT can potentially costimulate effects of these modalities for cancer treatment. In this respect, is worth focusing particular attention on the additional direct stimulatory effect of high fever range temperature, which is reproduced locally by hyperthermia, on immune cells [9]. Fever can systemically alert the immune system and enhances the efficacy of immune surveillance, both against infections and cancer. Short-term (30–60 min) heating at a high physiological fever range (39.5–41 °C) can promote antigen uptake and maturation of DCs, the activation of both CD4+ Th1-cells and CD8+ cytotoxic T-cells (CTLs), and CTL migration through high endothelial venules (HEV) compared to normal body temperature [83,84]. Cytotoxic activity and recruitment of NK cells are also promoted by fever-range hyperthermia through upregulating MICA (MHC class I polypeptide-related sequence A) on tumor cells and its receptor NKG2D on NK cells [85]. The same profiles of antigen-presenting costimulatory and Th1 cytokine response to DAMP signaling (see above) and of high-fever range temperature suggest that the damage signaling-related indirect and the direct immunomodulatory effects of moderate heat on immune cells may act in synergy. At 42 °C, tumor core temperature irreversible cell stress and systemic damage signaling dominates, while at about 40 °C, usually achieved in tumor adjacent normal tissues, immune cell functions may be supported, particularly if the treated region of pancreas, gastric, or breast cancer overlap with the spleen.

## 7. Combination with Dendritic Cell Immunotherapy

Antigen-presenting DC immunotherapy aims at improving immune response by isolation, loading with tumor antigen, and in vitro expansion of autologous DCs, and then reinjecting them into the body (DC vaccination) for elevated tumor-specific T cell activation and response [86]. The positive immunomodulatory effect of mEHT was expected to support DC therapy for a systemic antitumor effect, as was proven by two research groups. Tumor grafts of mouse head and neck squamous cell carcinoma cell line (SCCVII) were grown simultaneously in the left leg and in the chest of C3H/He mice. Only the left leg tumors were treated three times by repeated mEHT or intratumoral DC vaccination, or by combining these modalities [33]. Significantly increased apoptosis and reduced tumor sizes were detected progressively both in the mEHT and DC-treated groups compared to untreated controls. This was seen not only in the treated leg but also in the distant, untreated chest tumors. This systemic abscopal effect was further improved at both sites in the combined treatment groups. Elevated Hsp gp96 levels; DC activity; and increased numbers of CD3+, CD8+, and S100+ cells and reduced number of Foxp3+ cells were detected in the treatment-reactive tumors compared to untreated controls, which confirmed the ICD nature of the antitumor effect [75].

mEHT treatment also generated a favorable tumor microenvironment for DC therapy in directly treated single CT26 (C26) CRC grafts of BALB/c mice after DC vaccination [36]. Tumor sizes were progressively and significantly reduced during the 33 days follow up after both the mEHT alone and the mEHT combined DC therapy compared to DC treatment alone or untreated controls. Furthermore, mEHT plus DC injection prevented tumor seeding when the contralateral legs were re-challenged with the tumors. Elevated numbers of CD45+ leukocytes, F4/80+ macrophages, and eosinophil were counted along with enhanced release of Hsp70 and significant apoptosis, particularly in the most reactive treatment groups.

These data and those in the previous chapter suggest that under experimental conditions, in some but not all tumor types tested, mEHT can support the extension of its local, direct apoptosis induction into an ICD-mediated systemic tumor killing when combined with DC injection or immune stimulation.

## 8. Combination with Radiotherapy

Radiation therapy (RT) induces counter-effective hypoxia, which activates hypoxia-inducible factor-1a (HIF-1a) and its target genes, e.g., vascular endothelial growth factor (VEGF) [87]. On the contrary, loco-regional hyperthermia, including mEHT, can increase tumor perfusion and oxygenation, which promote the efficiency of RT-induced DNA breaks and inhibition of DNA repair mechanisms [88,89]. The mEHT treatment (41 °C for 30 min) immediately followed by 15 Gy (60Co) RT resulted in the most reduced growth of FSaII mouse fibrosarcoma allografts in C3H mice compared to mEHT or RT alone, or when these were combined in the opposite order [39,88,89]. Hypoxic areas labelled with pimonidazole were reduced so those of Hif-1a, CaIX (carboanhydrase IX), and Vegf levels, while blood perfusion assessed with Hoechst 33,342 as well as the apoptosis were increased more than with any other tested options.

After mEHT monotherapy, elevated tumor damage was also observed in 9L aggressive glioma, and MCF-7 (homozygous CDKN2Adel, heterozygous PIK3CA p.Glu545Lys, c.1633G > A; wild type TP53) hormone-sensitive breast cancer cell lines [42], as well as in Panc-1 (homozygous for TP53 gene mutation in c.818G > A, for CDKN2Adel; and heterozygous for KRAS c.35G > A [47]) pancreas ductal adenocarcinoma. Consecutive use of mEHT and RT resulted in additive tumor destruction both in radioresistant 9 L glioma and Panc1 pancreas adenocarcinoma cells compared to RT alone [42,47]. This combination therapy also inhibited the growth the tumor precursor/stem cell population, as demonstrated by the significantly reduced long-term tumor colony formation in both studies.

The electric field and heat induced by mEHT within 4 h prior to RT (repeated after 2 days) were also efficient radiosensitizers, increasing the SAR value of RT in lung adenocarcinoma cell lines A549 (homozygous KRAS p.Gly12Ser, c.34 G >A; STK11 p.Gln37Ter, c.109C > T mutations; and wild type TP53) and HT1299 (heterozygous NRAS p.Gln61Lys, c.181C > A mutation and homozygous TP53del) [44]. Gradual temperature elevation by mEHT resulted in the linear increase of the equivalent radiation dose of RT treatment. Combined treatment led to significant apoptosis and tumor volume reduction in vivo [44]. This was dependent on the dielectric properties of the tumors, and the measurement of dielectric features provided suitable temperature-mapping results [90]. These data suggested that clinical treatment planning could be improved by noninvasive measurement of patient-related tumor-specific dielectric properties using, for instance, magnetic resonance electrical properties tomography (MREPT) and dictionary-based electric properties tomography (dbEPT). These results confirm that the mEHT treatment can additively support the efficiency of RT-induced destruction of tumors of diverse genotypes and phenotypes.

## 9. Combination with Chemotherapy and Cancer Thermo-Sensitization

Tumor chemotherapy can also be improved in combination with mEHT, as has been confirmed by series of preclinical studies. While doxorubicin (Dox) treatment of C26 (also called CT26) CRC cell culture primarily led to tumor cell necrosis, mEHT monotherapy resulted in a massive apoptosis with significantly higher *γ*H2ax levels, indicating DNA DSBs, compared with after Dox [40]. Both treatments reduced long-term tumor colony formation after 10 days, indicating that they are also targeting the tumor progenitor/stem cell populations [91]. In combination therapy, mEHT promoted the uptake of Dox and had an additive effect on Dox-induced tumor destruction [29].

Liposomal doxorubicin (lipodox) treatment resulted in the most efficient in vitro killing of human hepatocellular (HepG2) and lung (A549) adenocarcinomas and U87-MG glioblastoma, as well as of mouse CT26 CRC, when combined with mEHT compared to WB heating either at 37 °C or at 42 °C [40]. Lipodox uptake measured by its fluorescence was facilitated by mEHT in vitro in the HerG2 > A549 > U87-MG > CT26 cell line order. Moreover, this was doubled in CT26 CRC grown in BALB/c mice compared to conventional heating at 42 °C. In line with this, mEHT combined with lipodox treatment resulted in the most significant tumor growth inhibition of the tested options in CT26 tumors in vivo. Major reduction in lipodox uptake by wortmannin after mEHT treatment indicated macropinocytosis as the main mechanism driving the liposomal drug engulfment by tumor cells [92].

Autophagy protects tumor cells against apoptosis that, therefore, can be overcome by autophagy inhibitors [93]. The autophagy inhibitor 3-methyladenine (3-MA) interfered with cellular damage recovery, promoted transitionally by mEHT treatment, in OVCAR-3 (homozygous TP53 *p*.Arg248Gln; c.743G > A mutation), a human high-grade ovarian serous carcinoma, and SNU-17, human papillomavirus (HPV) positive cervical squamous cell carcinoma cells [43]. Combined treatment induced more significant apoptosis and reduced tumor weight and volume in both xenografted tumors, so as in a patient-derived cervical cancer, more than monotherapy for any of the tested options. Increased phosphorylation at Thr180/Tyr182 of p38, a stress-dependent kinase, was detected in cancer cells, along with elevated cleaved/activated caspase-3 and PARP levels in OVCAR-3 and SNU-17 cells. Therefore, the apoptosis-inducing effect of mEHT treatment could be augmented by inhibiting autophagy, the pro-survival reaction that may also be supported initially by mEHT.

Efforts have also been made to stimulate heat absorbance of tumors compared to their microenvironment. Iron-dextran, incorporating Fe^3+^ ions with high dipole potential, was accumulated after intravenous injection in NCI-H460-luc2 large cell lung adenocarcinoma (homozygous KRAS p.Gln61His, c.183A > T and STK11 p.Gln37Ter, c.109C > T mutations and heterozygous PIK3CA p.Glu545Lys, c.1633G > A mutation) xenografts grown in BALB/c nude mice [45]. High intratumoral iron-dextran concentration served as a targeted thermosensitizer for mEHT-induced tumor destruction at the usual energy input. Selective temperature increase, up to 47 °C, could be achieved within the cancer tissue, which caused massive necrosis after mEHT (instead of the widely documented apoptosis at 42 °C), with only 40–42 °C measured in the adjacent normal tissues.

This was in contrast with the effect of 50 nm gold nanoparticles (AuNPs) on HepG2 hepatocellular carcinoma cultures, where extracellular AuNPs neither promoted mEHT-induced heat generation nor tumor cell killing, but instead protected cancer cells from damage, particularly when they were engulfed by tumor cells [46]. These findings are consistent with the importance of dipole character of mEHT targets resulting in strong interaction and heat generation with ionic iron but not with colloidal gold particles of only minor positive net-surface charge, which may even be shielded [28].

## 10. Clinical Utilization of mEHT for Upgrading Human Oncotherapy

Although intratumoral temperature measurement is not performed during clinical mEHT/oncothermia treatments, relevant published data show that the 150 W input power delivered using 13.56 MHz modulated radiofrequency can induce an appropriate temperature rise of 3–5 °C in deep tissues, simulating clinical treatment conditions. Studies in anesthetized living pigs of ≈50 kg demonstrated that the 150 W input power (above 90% of which was absorbed) and upper electrode size of 20 cm diameter, used also for human treatment combinations, can generate about 42 °C in pig livers within 15 min after starting the 60 min mEHT treatment, as deep intrahepatic temperatures were measured using fiber optic sensors [94]. This was also confirmed in a few elderly human patients with highly advanced cancers, where direct temperature measurements were similarly performed [28].

This loco-regionally delivered deep hyperthermia generated by mEMT has been an efficient and safe chemo- and radiosensitizer in human oncotherapy by improving both the local tumor control and survival rates, along with contributing to the systemic (abscopal) effect of ionizing radiation [10,95]. Besides many positive case studies being published in a wide range of tumors, several clinical papers analyzing the statistically relevant number of patients have come out recently confirming the added benefits of mEHT treatment to conventional oncotherapy. Combination therapies involving mEHT improved tumor response and survival to radio-chemotherapy in stage III-IV pancreatic cancer patients [96], the 5-year overall survival of astrocytoma from 24% to 83% compared to the best supportive care alone [97], the dose-dense temozolomide treatment of recurrent glioblastoma in a cost-effective way [98], or the local disease-free survival and local disease control in FIGO (International Federation of Gynecology and Obstetrics) stages IIB to IIIB squamous cell carcinoma of the cervix [99]. This latter phase III randomized controlled trial in advanced cervical cancer showed a high level of clinical evidence on the support of an abscopal effect by mEHT in combination with radio-chemotherapy [95]. The significant range of complete metabolic resolution (CMR) in extrapelvic lymph node metastases, observed in the mEHT combination treatment, is in agreement with the preclinical studies showing an oncoreductive effect of mEHT also in grafted tumors distant from the treated sites [12,33].

Randomized clinical trials have recently been running for testing the benefits of mEHT combined with folfirinox or gemcitabine chemotherapy in metastatic pancreatic cancers, weekly paclitaxel or cisplatin use in recurrent or persistent ovarian cancer, chemo-irradiation in locally advanced cervical cancer, and for improving quality of life in unresectable pancreatic cancer patients [10].

## 11. Modulated Electro-hyperthermia Induced Tumor Damage Mechanisms Revealed in Cancer Models

Although mEHT treatment is preferred to complement conventional oncotherapies, preclinical studies have revealed that it can induce tumor damage and inhibit tumor growth on its own through diverse downstream signaling pathways, as summarized in Figure 5. Briefly, the mEHT-delivered electric field can be enriched in cancer on the basis of its elevated oxidative glycolysis (Warburg effect), ion concentration, and conductivity compared to adjacent normal tissues [13,28]. This can provoke local heath, controlled at about 42 °C, and affect charged molecules both in the extracellular matrix and cell membrane lipid rafts where ion channels become leaky, leading to ion fluxes and disequilibrium [35,48].

Therefore, the effects of hyperthermia and the direct electric field might synergize at the nanorange of tumor cell membranes [23,100]. As a result, cell stress-related protective mechanisms involving chaperone molecules such as heat shock proteins and calreticulin were upregulated [11,12,29]. However, when mEHT was effective, these could not prevent DNA double-strand breaks [29,47], supported also by the treatment-related downregulation of PARP1 and perhaps BRCA2 DNA repair enzymes [37,43,69], as well as the activation of apoptosis pathways usually through both the extrinsic and intrinsic routes [12,29,47]. Stress- and death-related molecular heterogeneity induced by mEHT was subject to tumor type, but when the TP53 gene was intact, the dominant and general tumor damage mechanism was p53-mediated apoptosis [29,37,38,41], mainly but not exclusively following the caspase-dependent subroutine. Although less significantly, tumor growth inhibition by mEHT also involved the senescence pathway through the p53-induced upregulation of p21^waf1^ [41,47]. Apoptosis by mEHT additionally stimulated the upregulation, cell membrane translocation, and cell injury-associated release of DAMP signaling molecules, including the chaperones and HMGB1 [12]. These, by carrying tumor antigens and binding to their cognate receptors on antigen-presenting DCs, could promote the uptake, processing, and presentation of tumor antigens and could contribute to the elevated immune cell infiltration and secondary ICD-mediated tumor damage [75,101]. In combination therapies, mEHT treatment proved also to be a good chemosensitizer by enhancing drug uptake and tumor reductive effects [29,41], a good radiosensitizer by downregulating Hif1a and its target genes [39,42,44,47], and a good promoter of systemic immune response after immunstimulation or intratumoral antigen-presenting DC injection [33,36]. However, it must be emphasized again that the major mEHT-activated pathways and their extent of contribution depended on the type, likely to be determined by the genetic/epigenetic makeup, of the treated tumor.

## 12. Limitations of the Presented Studies

The limitations of the overviewed studies may include (1) the fact that not many research groups working in Japan, South Korea, Taiwan, Germany, and Hungary have performed and published preclinical studies on mEHT thus far. These groups, however, run independent projects led by their prime interests and approaches, with only occasional co-operations. (2) Thus far, apoptosis has been proven to be the major cancer death mechanism caused by mEHT; therefore, most groups focused on this and its relation to necrosis. However, other regulated cell death pathways including ferroptosis, autophagy, necroptosis, or parthanatos may also be induced by mEHT, which require further studies. (3) Although some preclinical studies showed a sizable tumor destruction effect of mEHT at lower temperature ranges of 38–40 °C linked to electric field [23,43] and dielectric properties of tumor tissues [90], most studies reviewed here used ≈42 °C. Since some clinical studies, best exemplified, for instance, in advanced cervical cancer [90,99,102], also showed lower peritumoral treatment temperatures (about 38.5–40 °C), the translation of preclinical results obtained at about 42 °C in clinical situations requires caution. A dominant effect of hyperthermia in this lower temperature range is likely to be enhanced perfusion and improved reoxygenation, which will result in strong radiosensitization and chemosensitization [103]. Higher temperatures, i.e., those closer to about 42 °C, can be more effective both in tumor control and overall survival [104,105], whose correlation has also been relevant in mEHT (oncothermia) treatment. To this end, an abscopal effect linked to mEHT use was revealed in the extrapelvic lypmh nodes [95] in one of the aforementioned cervical cancer cohorts by the Minnaar group [99]. This may either show an additional effect of mEHT electric field at lower intratumoral temperature (≈40 °C) or that the intratumoral temperature, wherein those in the tumor centers can even be higher, where preclinical studies at 42 °C always showed the most robust tumor damage spreading in time towards the tumor periphery [11,12].

## 13. Conclusions and Outlook

On the basis of the preclinical studies, mEHT can be oncoreductive either alone or in combination with chemo-, radiation-, and targeted therapy. A major promise of mEHT treatment lies in its induction of diverse scale, however, relevant tumor growth inhibition and damage in most tested tumor types exist, irrespective of their inherent epi-/genetic makeup. However, despite unlocking some molecular details of mEHT-related cancer damage in preclinical studies, tumor-selective biological markers that would predict mEHT efficiency still need to be found. Potential candidates for this to be studied can be the tumor intrinsic epi-/genetic features affecting stress, apoptosis, and immune-associated pathways, as well as those involved in fine-tuning of tumor metabolism, including oncometabolites, some of which affect electric conductivity.

The growing interest from oncologists in exploiting mEHT (oncothermia) in clinical combination therapies is expected to facilitate further research and phase studies, which are supported by the data and correlations summarized in this paper. Some preclinical mEHT data that can already offer chances for clinical translation are, for instance, the treatment-induced (1) DNA double-strand breaks and inhibition of PARP1 for combination with PARP inhibitors; (2) promotion of the release of immune stimulating tumor DAMP molecules, for combination therapy with immune stimulating cytokines, immune checkpoint inhibitors, or tumor primed DCs; (3) inhibition of hypoxia mediators and the resolution of radioresistance, for combination with radiotherapy, e.g., in radioresitant pancreas and glia tumors; or (4) induction of pro-apoptotic mediators, which can support chemotherapy or autophagy inhibitor-induced apoptosis.

Therefore, preclinical mEHT data can support rational treatment strategies for more efficient and personalized combinations of this locally targetable, non-invasive modality, with traditional chemo-, radiation-, or molecular targeted oncotherapy regimens.

## Figures and Tables

**Figure 1 ijms-21-06270-f001:**
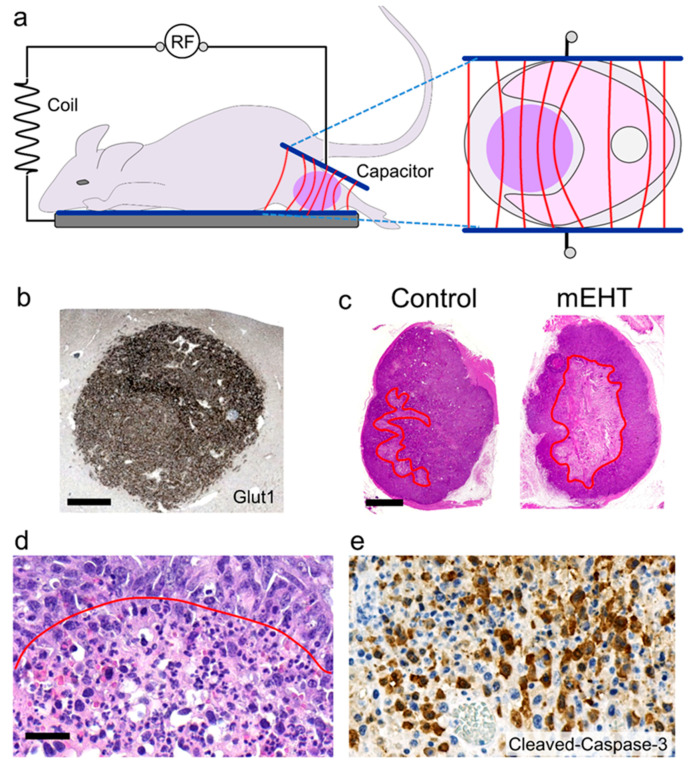
Treatment model, theoretical background, and major effects of 13.56 MHz amplitude-modulated radiofrequency (RF)-generated electric field (mEHT) on cancers grafted into mice (data from experiments by Vancsik et al., 2018 [12]). Enrichment of electric field (red lines) in the cancer lump (pink) (**a**) (this figure is modified from Meggyeshazi et al. 2014. [11]) due to elevated intratumoral glucose uptake indicated by high glucose transporter (Glut1)-level (brown) (**b**) glycolysis and ion concentration. Damaged (pale) tumor areas (encircled in red) measured and compared accurately on digital slides of untreated control and treated C26 allografts (**c**). Signs of massive apoptosis including nuclear shrinkage, chromatin condensation, and apoptotic bodies (**d**) (below red line) and accumulation of cleaved/activated caspase-3 positive tumor cells (brown) (**e**) after mEHT treatment. (**b**,**e**) DAB immunoperoxidase reactions; (**c**,**d**) hematoxylin–eosin staining. Scale bars, (b): 100 μm; (c): 400 μm; (d–e): 50 μm.

**Figure 2 ijms-21-06270-f002:**
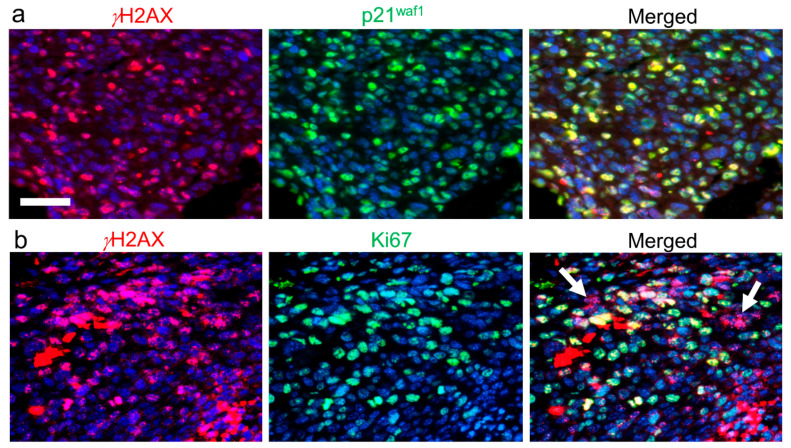
mEHT treatment-related DNA double-strand breaks indicated by histone 2AX (H2ax) phosphorylation (*γ*H2ax, red) in a C26 colorectal cancer allograft (unpublished data from the experiments of Vancsik et al., 2018 [12]). High *γ*H2ax levels: (1) inducing the upregulation of p21^waf1^ cyclin-dependent kinase inhibitor (green) protein ((**a**) see their complete overlap); (2) the non-proliferating, Ki67 (green)-negative tumor cells (arrows) are clearly related to mEHT treatment (**b**). In terms of immunofluorescence double labeling, cell nuclei are stained using 4’,6-diamidino-2-phenylindole (DAPI) (blue). Scale bar: 50 μm (for all).

**Figure 3 ijms-21-06270-f003:**
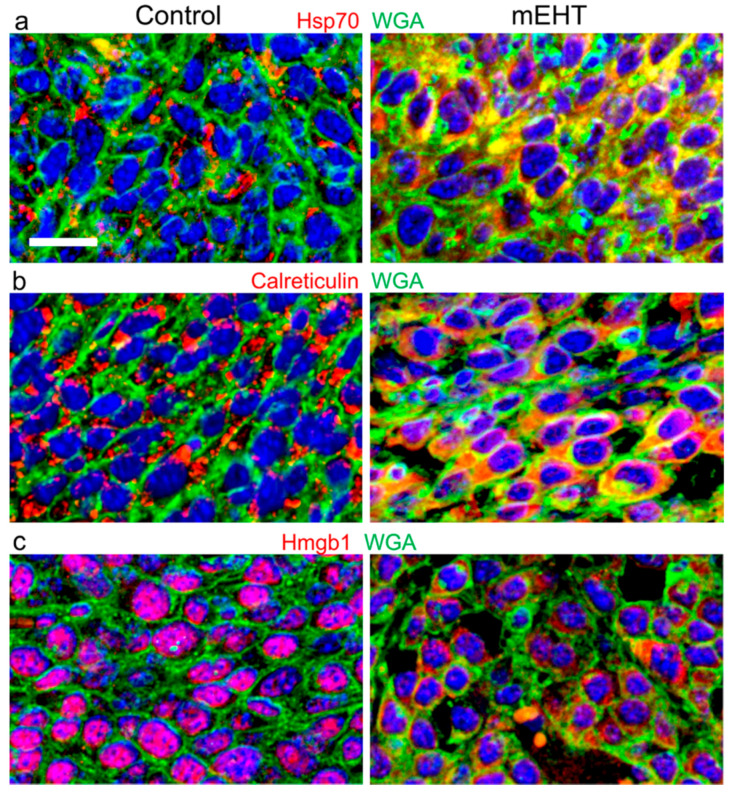
mEHT-induced upregulation and cell membrane (labelled with fluorescein isothiocytante (FITC)-wheat germ agglutinin (WGA), green) translocation of calreticulin (red; their co-localization is yellow) from the endoplasmic reticulum after 12 h (**a**) and Hsp70 (red) from its cytoplasmic storage vesicles after 48 h (**b**) (data from experiments by Vancsik et al., 2018 [12]). Nuclear-to-cytoplasmic translocation, then extracellular release and loss from cells of Hmgb1 after 48 h post-treatment (red on (**c**)). C26 colorectal cancer (CRC) allografts, immunofluorescence double labeling, cell nuclei are blue (DAPI). WGA: wheat germ agglutinin, lectin. Scale bar: 20 µm (for all).

**Figure 4 ijms-21-06270-f004:**
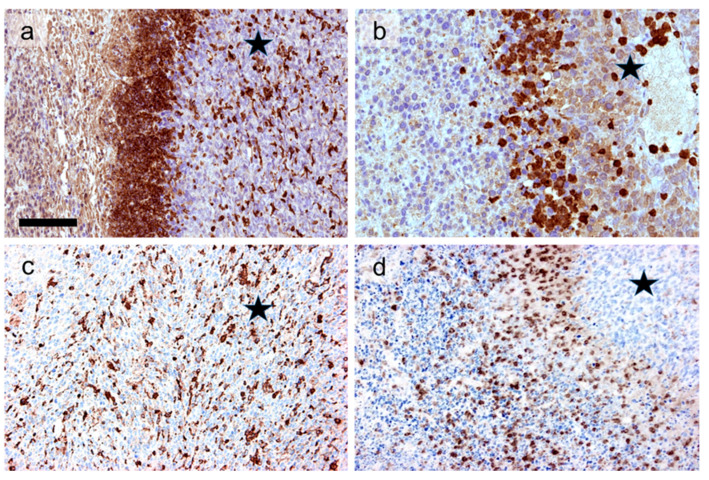
mEHT treatment-induced tumor damage and accumulation of CD3+ T cells (**a**), S100-positive antigen-presenting dendritic-cells (**b**), granzyme B-positive NK cells (and T cells together) (**c**), and f4/80-positive macrophages (**d**) (data from experiments by Vancsik et al., 2018 [12]). Asterisks label intact-looking tumor regions opposite to the apoptotic left sides of the images. The almost missing macrophages from the intact-looking tumor region ((**d**), asterisk) suggest the secondary involvement of this cell type in mEHT effect. DAB: immunoperoxidase reactions (brown chromogen). Scale bar is 100 μm for (**a**,**c**,**d**), and 50 μm for (**b**).

**Figure 5 ijms-21-06270-f005:**
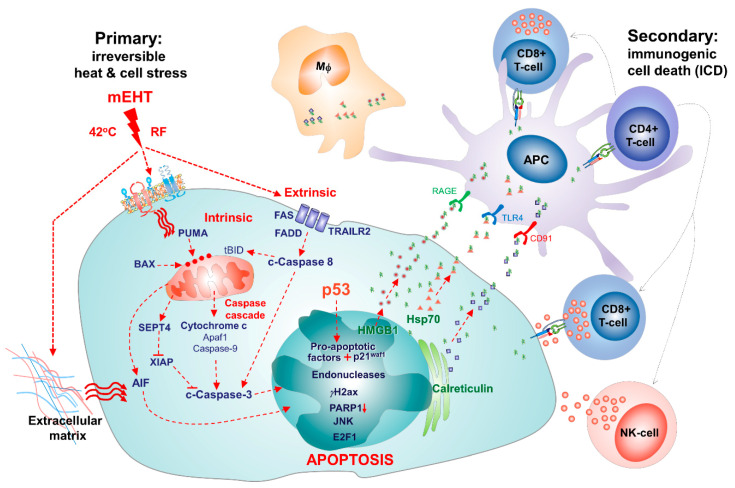
Summary of major tumor destruction related effects of mEHT in tumor models. The primary impact of the radiofrequency-induced (RF) electric field and 42 °C heath affects dipole cell membrane molecules including death receptors concentrated in lipid rafts and charged molecules in the extracellular matrix. mEHT-induced cell stress activates apoptotic pathways, leading to elevated expression and membrane translocation of damage-signaling molecules. Release of damage signals may promote the uptake of tumor antigens and activate adaptive and innate immune cells for a secondary immunogenic cell death (ICD), as revealed in mouse C26 (CT26) colorectal adenocarcinoma and squamous cell carcinoma cell line (SCCVII) head and neck squamous cell carcinoma models. (Upgraded from Vancsik et al., 2018 [12]).

**Table 1 ijms-21-06270-t001:** Antibodies and conditions of their use for immunoperoxidase and immunofluorescence.

Antibody	Reference No.	Dilution	Antigen Retrieval	Vendor
Calreticulin	#12238	1:200	T-E	Cell Signaling
CD3	#IS503	(RTU)1:2	T-E	Dako
*F4/80	#16-4801-86	1:100	T-E	Thermo
GLUT1	#355A-14	1:100	T-E	Cell Marque
H2AXγ	#9718	1:50	T-E	Cell Signaling
HMGB1	#6893	1:200	Citrate	Cell Signaling
HSP70	#4872	1:50	T-E	Cell Signaling
Ki67	#MA5-14520	1:400	T-E	Thermo
p21^waf1^	#MA5-14949	1:50	T-E	Thermo
S100	#RB-9018	1:500	T-E	Thermo

**Table 2 ijms-21-06270-t002:** Effects and main mechanisms of action of capacitive coupled 13.56 MHz radiofrequency using 1/*f* amplitude modulation in published tumor models.

Tumor Type (Mice)	Treatment/Model	Major mEHT Effect	Main Mechanism	Publication
HT29 human CRC xenograft (BALB/c nude mice)	single mEHT shot, 42 °C, 30 min; compared with infrared heating to 42 °C	≈3 times higher tumor damage than after radiation heating	synergy between the temperature-dependent (heat) and -independent (electromagnetic field) effects	Andocs et al., 2009 [23]
HT29 human CRC xenograft (BALB/c nude mice)	single mEHT shot, 42 °C, 30 min	significant tumor apoptosis	extrinsic pathway activation; up-regulation of TRAIL-R2, FAS, FADD	Meggyeshazi et al., 2013 [32]
HT29 human CRC xenograft (BALB/c nude mice)	single mEHT shot, 42 °C, 30 min	significant tumor apoptosis	DNA fragmentation; AIF1-mediated apoptosis	Meggyeshazi et al., 2014 [11]
SCCVII mouse head and neck squamous cell carcinoma (C3H/He mice)	intratumoral DC injection + 3 × mEHT: 9, 11, and 13 days after inoculation	reduced tumor sizes, both at the treated and distant locations	elevated DC activity, Hsp gp96 levels, CD3+ and CD8+, and S100+ cells; reduced Foxp3+ cells	Qin et al., 2014 [33]
HT29 human CRC xenograft (BALB/c -nu/nu mice)	single mEHT shot, 42 °C, 30 min	significant cell stress and apoptosis	upregulation and cell membrane translocation of DAMP signals (HSP70, HSP90, calreticulin, HMGB1)	Andocs et al., 2015 [34]
U937 human myelomonocytic lymphoma cells	single mEHT shot, 39–46 °C vs. WB, 30 min.; in vitro and in silico modeling of mEHT on membrane lipid rafts	similar apoptotic cell death with mEHT at 39 °C to that of WB at 44 °C	selective energy absorption (hot spots) focused on membrane rafts	Andocs et al., 2015 [35]
CT26 mouse CRC allograft (BALB/c immunocompetent mice)	single mEHT shot, 42 °C, 30 min + intratumoral injection of DCs	significant apoptosis; additive effect of mEHT on DC therapy	Hsp70 release and elevated cytotoxic T cell number and activity; prevention of tumor seeding after tumor re-challenge	Tsang et al., 2015 [36]
U87-MG and A172 human glioma cells (BALB/c nude mice)	3 × mEHT, 42 °C, 60 min (every other day) in vitro and in vivo	significant apoptotic cell death; reduced tumor cell migration	increased E2F1 and p53, reduced PARP1 mRNA levels; reduced proportion of CD133+ stem cell fraction	Cha et al., 2015 [37]
U937 human myelomonocytic lymphoma cells	single mEHT shot, 42 °C, 30 min; comparison with WB heating, in vitro	significant apoptotic cell death (mainly protective effect of WB)	caspase-mediated apoptosis; FAS, c-JUN N-terminal kinases (JNK), and ERK signaling upregulation	Andocs et al., 2016 [18]
Human Huh7 hepatocellular cc. and HepG2 hepatoblastoma cells (BALB/c nude mice)	3 × mEHT, 42 °C, 60 min (every other day) in vitro and in vivo	significant apoptotic cell death in both cell types in vitro; growth inhibition of HepG2 in vivo	suppressed cell proliferation and long term colony formation; upregulation of septin-4, p53, and p21^waf1^	Jeon et al., 2016 [38]
HepG2 human hepatoblastoma cells	single mEHT shot, compared to capacitive coupling HT (cCHT) and WB heating at 42 °C, 30 min, in vitro	similar range of apoptosis by mEHT at 42 °C to that by WB at 46 °C	HSP70 upregulation by all three treatments; caspase-dependent apoptosis only by mEHT	Yang et al., 2016 [24]
C26 mouse CRC allograft (BALB/c immunocompetent mice)	single mEHT shot, 42 °C, 30 min	significant apoptotic tumor damage	blockade of cell cycle progression; caspase-mediated apoptosis, DAMP signaling, ICD	Vancsik et al., 2018 [12]
FSaII mouse fibrosarcoma allograft (C3H mice)	single or 3 × mEHT shot(s), 41 °C, 30 min + RT 15 Gy ^60^Co irradiation	enhanced tumor apoptosis and reduced tumor growth by mEHT; additive effect of mEHT on RT apoptosis	increased blood perfusion and tumor oxygenation; reduced Hif-1a, CaIX, and Vegf levels	Kim et al., 2018 [39]
C26 mouse CRC cells	single mEHT shot, 42 °C, 30 min, in vitro	significant cell stress and apoptosis; additive effect on doxorubicin	upregulated γH2ax and p-p53(Ser15), and downregulated p-Akt (Ser473); inhibited long-term colony formation	Vancsik et al., 2019 [29]
HepG2 human hepatoblastoma, A549 (lung cc), and U-87MG (glioblastoma) cells, and CT26 mouse CRC cells (BALB/c mice)	liposomal doxorubicin (Lipodox) + single mEHT shot, 42 °C, 30 min, in vivo (CT26 only) and in vitro (all cell lines)	mEHT significantly increased Lipodox (and 70 kDa dextran-FITC) uptake in HerG2 >A549 >U87MG >CT26; Lipodox + mEHT: most significant tumor (CT26) reduction in vivo	macropinocytosis indicated by the prevention of Lipodox uptake (and mEHT effect) by wortmannin	Tsang et al., 2019 [40]
B16F10 melanoma cell culture and allograft (C57Bl/6 mice)	3 × mEHT, 42 °C, 30 min; 4, 6, and 8 days after inoculation, in vivo and in vitro	reduced tumor size and induction of γH2AX	upregulation and release of DAMPs (Hsp70, Hmgb1, ATP); elevated p53, p21^waf1^, and p27^kip1^ (senescence) and NK cell number; reduced Mhc-I and melan-A	Besztercei et al., 2019 [41]
9 L human gliosarcoma, MCF-7 (breast cc), and MDKC (canine kidney epithelial cells)	single mEHT shot, 42 °C, 30 min + RT 10 MV 5 gyirradiation in vitro	supra additive tumor damage in L9 radioresistant glioma cells	significantly reduced tumor precursor cell fraction in IL9 and MCF-7 by clonogenic assay	McDonald et al., 2018 [42]
Human ovarian (OVCAR-3, and SK-OV-3) and cervical (HeLa and SNU-17) cancer cell lines (BALB/c nude mice)	single mEHT shot, 42 °C, 30 min + autophagy inhibitor 3-methyladenine (3-MA); in vitro and in vivo	significant apoptosis (increased sub-G1-phase fraction) reduced tumor xenograft weight and volume; combined treatment caused additive tumor damage	phosphorylation of p38, elevated caspase-3, and PARP, and mEHT-induced cellular damage recovery (autophagy)	Yang et al., 2019 [43]
A549 and NCI-H1299 human lung adenocarcinoma cell lines (BALB/c nude mice)	2 x mEHT shot, 42 °C, 30 min + 2 x RT 2–8 Gy; in vitro and in vivo	significant radiosensitizing effect, apoptosis, and tumor volume reduction	increased equivalent radiation dose by mEHT; dependence on tumor dielectric properties	Prasad et al., 2019 [44]
NCI-H460-luc2 human lung adenocarcinoma cell line (BALB/c nude mice)	mEHT every third day for 5 weeks, combined with i.v. iron-dextran solution (mEHT + IronD) in vitro and in vivo	significantly higher tumor necrosis in mEHT + IronD than after paclitaxel monotherapy	mEHT + IronD increased tumor temperature to 47°C compared to 42 °C after mEHT monotherapy, resulting in increased tumor sensitization	Chung et al., 2019 [45]
HepG2 human hepatoblastoma cell line	mEHT, 15W for 10 min plus 50 nm size spherical-, urchin-, or rod-shape gold nanoparticles (AuNP) in vitro	AuNP in the medium: no tumor damage; cell-incorporated AuNP: tumor protection	AuNPs absorbed mEHT energy at this power without temperature increase, independent of particle shape	Chen et al., 2019 [46]
Panc1 human pancreas ductal adenocarecinoma cell line	single mEHT shot, 42 °C, 60 min, combined with RT (Cs-137/2 Gy) in vitro	significant apoptosis in mEHT + RT combination, resolved tumor radioresistance	upregulated caspase-3 and p21^waf1^, reduced p-Akt levels; DNA double-strand breaks	Forika et al., 2020 [47]
HT29 and SW480 human colorectal adenocarcinoma cell lines	mEHT 42 °C, 60 min, in comparison with WB heating; in vitro	mEHT induced significantly reduced proliferation and clonogenicity	induction of ion fluxes through tumor cell membrane channels; disequilibrium of most ions; tumor damage	Wust et al., 2020 [48]

mEHT: modulated electro-hyperthermia; CRC: colorectal cancer; TRAIL-R2: tumor necrosis factor-related apoptosis-inducing ligand receptor type-2; FAS: tumor necrosis factor receptor superfamily, member 6; FADD: Fas-associated protein with death domain; DAMP: damage associated molecular pattern; WB: water bath; DC: antigen-presenting dendritic cell; AIF1: apoptosis-inducing factor; PARP: poly-adenyl ribose polymerase; ICD: immunogenic cell death; RT: radiotherapy; Hif-1a: hypoxia inducing factor-1alpha; CaIX_carboanhydrase-9; Vegf: vascular endothelial growth factor.

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
