# Peer review of "Modulated Electro-Hyperthermia-Induced Tumor Damage Mechanisms Revealed in Cancer Models"

_ijms, 2020, doi:10.3390/ijms21176270_

Round 1
Reviewer 1 Report
Review Comments:
The authors have represented a review article entitled “Hyperthermia induced cell stress, apoptosis, damage signaling and support of immunogenic cell death in cancer models after modulated electro-hyperthermia treatment”. The aims and objectives of the manuscript is good. The review is in early stage. This study needs some revision before accepted for publication.
Comments:
- Please try to explain the status of this review compared to presently reported literature.
- Please rename the title of the review. The title is not specific and the authors included too many keywords. Please represents one compact brief article title.
- Please emphasize more clearly what is the importance of this review? How researchers will be benefitted?
- Please remove supplementary section and put the matter in your main text. Review article itself is a collection of data, there is no need for separate supplementary section.
- Make a paragraph and clearly explain how SAR value is calculated.
- In Table 1, please add one more column which will define what materials or nanoparticle used for this study.
- Please rewrite the section 3.2., with more scientific input. Authors just represent the data as it is.
- In section 11, Limitations section may be removed and rewrite in another section which will only reflects the Limitations.
- Make a separate section “Mechanism of Hyperthermia” and elaborate.
- Conclusion section must be re-written with a proper concluding remarks not in single line but with a sense of knowledge regarding the principle of hyperthermia and their achievement in field of cancer treatment..
Author Response
Below, please find our point-by-point comments and corrections (which are highlighted in red in the revised manuscript) made based on your reviewers’ criticisms:
To reviewer 1.
1-3. Our manuscript is substantially reviewed in the abstract and in the finishing parts by including a “Modulated electro-hyperthermia induced tumor damage mechanisms revealed in cancer models” and a “Conclusion” section (and a separate “Limitations…” section) where we further emphasized the unique features of this special way of electrohyperthermia compared to other hyperthermia techniques. The title was also mad more focused to “Modulated electro-hyperthermia induced tumor damage mechanisms revealed in cancer models”. Also, we included some additional references including 2 recently published papers by Wust et al. and by Forika et al. with oncotherapy focus and 3 papers (2 by Gas et al. and one by Papp et al.) on the biophysical features of electrohyperthermia in cancer (suggested by Revier 2), which were not reviewed yet in the original manuscript. We further emphasized that our paper is especially focused on mEHT, and that this systematic review is summing up and trying to synthetize the preclinical data of this technique for supporting clinical utilization.
Also, a complex keyword “ DNA double-strand breaks” is eliminated from R1.
The importance of this review is clarified and emphasized more in the abstract and in the conclusion parts in the revised manuscript.
- The whole supplementary part including its table (which became Table 1., thus the original Table 1. became Table 2.) of the original manuscript became embedded as a sub-section of the “2. Standardized mEHT Treatment and Protocols” as “2.1 Methods Used for Demonstration Images”.
- SAR calculation in mEHT is explained in details so as “pink noise” in modulation, in the revised manuscript.
- The original Table 1 (now Table 2.) summarizes amplitude modulated electro-hyperthermia treatment related effect we added 4 additional relevant papers into this studying Fe3+ (Chang et al. 2019), or gold nanoparticles (chen et al. 2019), a most recent one on the non-thermal effects of mEHT (Wust et al. 2020), and also a very recent paper in IJMS on mEHT combined with radiotherapy by Forika et al. (2020).
- We added some sentences on the importance of p53 mediate pathways to introduce section 3.1 and rewrote the introduction of 3.2 to put mEHT induced p53 independent apoptosis into context. Also, we inserted some explanatory definitions into the results presented in 3.2.
- The “Limitations” part was separated to become section 11., we introduced a separate section 12. On the “Mechanism of Electro-hyperthermia induced damage…” based on the original “Summary” and separated a “Conclusion as and Outlook” part, which became section number 13th where we specified how some preclinical observations may be translated into clinical applications, as suggested by the reviewer.
- A separate section “Mechanism of Hyperthermia” was elaborated (see also above) linked to Fig. 5. which figure was also corrected at “42oC” (space was erased), gamma is written in italic.
- In a separate and final “Conclusion” part 13th (see also above) focused on emphasizing some specific examples on the potential clinical translation of the preclinical results.
To reviewer 2.
Literature citation style was modified into the IJMS template style and corrected the indicated mistakes accordingly.
Page numbers are corrected in original Ref. [89].
The mistaken sentence related to Ref [62], and the criticized 2nd sentence from “Clinical utilization”… are erased in revised R1.
Figure 5 was also corrected: the space was eliminated at 42oC and NK-cell membrane was exchanged to darker line to be recognized.
The ExPasy database website is moved to the references.
We substantially revised the abstract and the final “Mechanism… and “Conclusion …” parts of the manuscript part of the original manuscript to put the purpose and potential clinical translation of this review into context. See details in our answers to reviewer 1 (above) and in the revised manuscript.
PubMed is the most widely accepted indexing of medical oncology, which we focused on and reviewed only preclinical models of using high-fever range (~42 oC) temperature heating and not to ablation. However, since arXiv:1710.00652 paper adds important historical overview of hyperthermia as a whole, and doi:10.24425/aee.2019.129339 paper adds crucial physical backround of electrohyperthermia in tissues we included and refer to both of these papers added with a 3rd one by Papp et al 2017 (not in PubMed) on the biophysical approach in our R1 version. Also, 2 additional recent papers were added by Wust et al. and by Forika et al. see also our answers for reviewer 1.
Criticism: …”at least 16 of the 99 referred papers are directly or indirectly related to the authors of this review.” Yes, indeed, this background of our substantial experience in this field prompted us to overview the area and to produce this review. This approach is hopefully more scientific that writing a review by someone not embedded into this special field.
Publications in Table 1 (now Table 2.) are linked with their appropriate references.
DC as “antigen presenting dendritic cells” is written out fully in the abstract of R1. When used in Table 1. (now Table 2) it is explained at the end of this table, and in its first appearance also in the body text on page 10 of R1.
Concerning symbols used in “pink noise” and definition of SAR we explained them in more details in R1. Please also see or response to reviewer 1 above.
Concerning the demonstration figures the major technical details how they were produced are included into the body text from the Supplementary part where the published papers related to the experiments of Figs 1-5 are also referred to (including the sources of the original drawings).
We corrected the name of melanoma cell line to B16F10.
Furthermore, we corrected the listed editorial mistakes, including lowercase letters for keywords; citation formats were corrected to the IJMS usual format involving multiple citation formats in the text; unnecessary spaces are removed, so as between numbers and temperature signs; initial letters in heading words were changed into uppercase; the terms ‘in vivo’, ‘in vitro’ now are written in italic style, so as variables such as 1/f, or Greek symbols; in abbreviations ‘et al.’ missing dots are added, so as missing spaces inserted between numbers and their related units.
Many thanks!
Tibor Krenacs
Reviewer 2 Report
The loco-regional hyperthermia inhibits DNA repair enzymes, increases the flow of oxygen to hypoxic tumor cells, and makes them more sensitive to radiotherapy. Additionally, by increasing blood flow, it supports chemotherapy and the delivery of cytostatics to the cancerous tissue. The modulated electro-hyperthermia (mEHT) with a frequency of 13.56 MHz (so-called oncothermia) has lately been an emerging way of delivering loco-regional heating with favorable safety and tolerance profiles in clinical practice.
The authors overview the published preclinical data articles from recent years on the molecular mechanisms of damage in mEHT treated tumors with a frequency of 13.56 MHz and the correlations between different mEHT treatments and their effects in the treatment of various neoplastic lesions. However, they only limited themselves to PubMed database by searching for the terms ‘oncothermia’, ‘electrohyperthermia’, or ‘electro-hyperthermia’ linked or not with the phrase ‘modulated’. Presented studies reveal important pathways and mechanisms, which deserve interest and may help improving rational treatment strategies for more efficient and personalized combination therapiess of mEHT with traditional chemo-, radiation-, immuno- or molecular targeted oncotherapy regimens.
In my opinion the obtained results are interesting and may be worth publishing in this journal after major revision.
The manuscript is written with proper English and punctuation.
Literature is mostly adequate but all references should be rewritten according to the IJMS template style:
https://www.mdpi.com/journal/ijms/instructions
In references list all authors names (do not use abbreviation et al.). In refs. [44,87,89–91,96–99] authors names in missing. Remove the DOI numer in [86]. In [89] check page numbers.
Journal names should be written in italic styles, publication year in bold and publication volume in italic styles.
I do not know what is the relationship of the reference [62] with the presented manuscript?
I do not find presented content in page 15 for references [89] and [13].
The ExPasy database website given in the manuscript text on page 3 should be moved to the references.
Authors should clearly state the purpose of the work. What is its message for readers. Why does the presented review have the character of 'briefly overview' and why does such a publication make any sense at all? What does this overview contribute to the discussed method/discipline/field? Why is such review better than individual papers? Why did the authors limit references to publications from PubMed and not include papers indexed in Web of Science or Scopus? Why only new publications were presented and the historical background of the method (papers write before 2005) is not given? What was the selection key for the publications gathered in Table 1? Why most of them are from research centres directly related to the authors of the manuscript? I recognised at least 16 of the 99 referred papers are directly or indirectly related to the authors of this review. Isn't that art for art's sake? In my opinion, the paper needs to be enlarged by others papers. I strongly recommend referring the following papers: arXiv:1710.00652 and doi: 10.24425/aee.2019.129339.
Link the publications from Table 1 with appropriate references.
The abbreviation ‘DC’ appears very frequently in the manuscript. The term: 'DC injection' firstly has been used in the abstract and is not explained. Then in the manuscript text appear statements: 'DC activity', 'DC therapy', ‘DC immunotherapy',' DC vaccination', ‘DC treatment'. Does the DC symbol means ‘dendritic cell’ as indicated under Table 1 and in page 10?
In page 3 the authors have written: ‘Amplitude modulation of electromagnetic waves using 1/f “pink noise” increases SAR’. What do symbols f, 1/f, SAR mean? What is 1/f amplitude modulation and how it works? Explain this or cite the proper literature.
The quality of drawings is good enough.
Authors should explain whether the drawings used in Figures 1–5 are made by the authors and have never been published in other journals. I found similar drawings in references [9,10]. If these pictures are copied/modified from other publications, please provide appropriate references and a note regarding copyright in figure captions.
In line 247, the M16F10 cells probably erroneously given instead B16F10.
Additionally, there are some editorial mistakes in the manuscript text:
In Keywords section use lowercase letters.
For multiple citations and numerical ranges use long dash, e.g. [3–4], 7–12 days, etc.
Remove spaces between listed references, e.g. [1,2].
Remove spaces between numer and temperature unit, e.g. 42°C.
First letters in headings should be capitalized.
Terms like: ‘in vivo’, ‘in vitro’ should be written in italic style.
All variables should be written in italic styles, e.g. 1/f.
In abbreviations ‘et al.’ add missing dots.
Greek symbols should be written in italic styles, e.g. γ-irradiation, γ-H2AX.
Add missing spaces before units, e.g. 10 MV, 5 Gy.
Detailed remarks and suggested corrections are included in the attached pdf file with notes.
Please, answer the reviewer's comments in details and mark all changes in the manuscript text.
Summarizing, the manuscript may be interesting but it needs some improvements before it will be accepted for printing in this journal. In my opinion, the manuscript may deserves to be published in the International Journal of Molecular Sciences after the major revision and author's corrections taking into account the recommendations indicated by the reviewers.

Author Response

(The authors gave the same response as above.)

Round 2
Reviewer 2 Report
The authors overview the published preclinical data articles from recent years on the molecular mechanisms of damage in mEHT treated tumors with a frequency of 13.56 MHz and the correlations between different mEHT treatments and their effects in the treatment of various neoplastic lesions.
In my opinion the obtained results are interesting and may be worth publishing in this journal after minor revision.
The presented manuscript blurs the border between a review article and a research article. I am not aware of reviews whose authors present the results of their previously published research to such a wide extent. I identified that over 20 publications referred are related directly or indirectly to the authors of the manuscript (Hungarian centre). In my opinion, if the article had new original research results, it could be successfully published as a research article. On the other hand, the authors' contribution to the creation of the article should be appreciated, although its meaning is not entirely clear to me.
Some references should be rewritten according to the IJMS template style.
References [12] and [30] are doubled. Reference [30] should be removed and the other references should be renumbered.
I do not know what is the relationship of the reference [80] with the presented manuscript?
Please, check the SAR unit on page 3.
All figures should be moved to pages after directly referenced.
Detailed remarks and suggested corrections are included in the attached pdf file with notes.
Please, answer the reviewer's comments in details and mark all changes in the manuscript text.
Summarizing, the manuscript should be published in the International Journal of Molecular Sciences after minor revision and author's corrections taking into account the recommendations indicated by the reviewers.

Author Response
We corrected all the points Reviewer 2. raised, including:
References are corrected to the IJMS template style.
References [30] was erased and replaced to [12] because it was a copy of ref [12].
The mistakenly included reference 80] was corrected to an appropriate one.
kJ is used instead of W/kg because the time factor is also involved in mEHT SAR definition.
Figures are relocated as close as possible to their first mention in the text.
All remarks indicated have been corrected in the revised R2 version.